# Anatomical Evaluation of Posterior Maxillary Roots in Relation to the Maxillary Sinus Floor in a Saudi Sub-Population: A Cross-Sectional Cone-Beam Computed Tomography Study

**DOI:** 10.3390/healthcare11010150

**Published:** 2023-01-03

**Authors:** Abdulaziz Abdulwahed, Mohammed Mustafa, Mohmed Isaqali Karobari, Ahmad Alomran, Khalid Alasimi, Abdulrahman Alsayeg, Abdullah Alsakaker, Hadi Mohammed Alamri

**Affiliations:** 1Department of Conservative Dental Sciences, College of Dentistry, Prince Sattam Bin Abdulaziz University, P.O. Box 173, Al-Kharj 11942, Saudi Arabia; 2Conservative Dentistry Unit, School of Dental Sciences, Universiti Sains Malaysia, Health Campus, Kubang, Kerian 16150, Kelantan, Malaysia; 3Department of Conservative Dentistry & Endodontics, Saveetha Dental College and Hospitals, Saveetha Institute of Medical and Technical Sciences, Chennai 600077, India; 4Dental Intern, College of Dentistry, Prince Sattam Bin Abdulaziz University, P.O. Box 173, Al-Kharj 11942, Saudi Arabia; 5Consultant Endodontist, Department of Endodontics, Prince Abdulrahman Advanced Dental Institute, Ministry of Defence, Riyadh 11564, Saudi Arabia

**Keywords:** endodontics, endodontic microsurgical planning, maxillary sinus floor, posterior maxillary roots, root canal treatment, Saudi Arabia

## Abstract

To evaluate the mean distance and differences between posterior maxillary teeth and maxillary sinus floor (MSF) concerning the age and gender of the patients, a total of 124 maxillary sinuses and 496 posterior maxillary teeth were randomly selected in 62 cone-beam computed tomography (CBCT) images. Mean distances between posterior maxillary roots (PMRs) from different teeth and the MSF were measured using a calibrated tool in the software. Other relations regarding the gender and age of the patients were determined. The mean root–MSF distances in the right and left first premolars were more significant compared to the second premolars. No significant relation was found between the apices of the right and left first and second molar roots and the floor of the maxillary sinus concerning gender. A statistically significant relation was found between the apices of the buccal root of the right first premolar, right and left first and second molars and floor of the maxillary sinus concerning the age group 21–40 years (*p*-value = 0.009). This study showed that the second molar mesiobuccal root apex is frequently related to the sinus floor. Differences were reported concerning age, concerning the distance between posterior maxillary teeth and the maxillary sinus floor. CBCT technology helped provide the clinical proximity between the MSF and the posterior teeth root apices during the treatment planning.

## 1. Introduction

The maxillary sinuses are air cavities inside the maxilla that are connected to the nasal cavity via the ostium [1]. Sinus is coated with a thin membrane called the Schneiderian membrane and is attached to the periosteum. Its thickness is estimated to be around 1 mm; hence, this membrane is not visible in the radiograph unless it becomes infected or irritated by an allergic reaction. Allergies or infections may cause an increase in the thickness of the membrane, resulting in radiographic detection. When the thickness of the Schneiderian membrane is over 3 mm, it is mainly considered pathological [1]. Throughout the intrauterine life, maxillary sinuses begin developing and continue until the child’s birth [2]. Along the maxillary arch, it is known that the maxillary sinus floor (MSF) and the root apices of posterior teeth have a closer relationship compared to its relationship with the root apices of anterior teeth [3].

Previous research has shown that in around 50% of the patients, the MSF might extend to the alveolar bone and create alveolar crypts that stretch between adjacent teeth or roots [4]. These anatomical associations have been investigated to assess the spread of the odontogenic infection from maxillary molars to its adjacent MSF [5]. Approximately 10–12% of all sinusitis patients were confirmed to have odontogenic infections that cause maxillary sinusitis [6]. In the dental literature, the pathogenic role of dental problems in developing sinusitis in the maxillary sinus is well described [7]. Maillet et al. examined 82 CBCT scans of patients displaying maxillary sinusitis and found that more than 50% had a dental etiological factor [8]. The height of the alveolar ridge decreases with age and the distance between the posterior maxillary roots (PMRs) and the MSF. Applying endodontic instruments, intracanal medicaments, and root-filling material into the maxillary sinus results in an inflammatory reaction of sinus mucosa [9]. This may result in sinusitis and sinus membrane thickening.

Moreover, mucosal thickening might also occur due to perforation during endodontic microsurgery [3]. In addition, other surgical interventions, such as tooth extraction or implant placement, may lead to sinusitis, oroantral fistula, and root displacement [10]. The best way to prevent these complications is to know the relationship between the MSF and PMRs [4]. Additionally, the relationship’s topography between the MSF and PMRs is a significant factor in orthodontic therapy’s motion and prognosis [2,5].

The maxillary sinus is closely related to the alveolar crest and can also be perforated by the teeth’ apices [1]. The patient’s age, the size of the maxillary sinus, and the degree of its pneumatization may influence the vertical relationship between MSF and posterior maxillary roots [11]. Proper region imaging is the easiest way to determine the relationship between MSF and PMRs. Periapical radiographs, orthopantomography, and cone-beam computed tomography (CBCT) are widely used dental imaging techniques. CBCT offers geometrical representations that prevent the superimposition of neighboring objects [4]. Further, CBCT is used to assess endodontic procedures and is expected to measure maxillary sinuses since both bone and soft tissue may be examined with thin sections in different views [6].

CBCT facilitates diagnosis and provides clinicians with a 3D knowledge of anatomy and detecting aberrant anatomical structures. Axial, sagittal, and coronal images produced by CBCT allow the clinician to consider the whole anatomic tissue structure [6]. There is a scarcity of research about this topic, especially in the Saudi population. Some case reports have reported the effects of overinstrumentation, overfilling, and the extrusion of endodontic materials or irrigants into the maxillary sinus [12,13,14]. These effects may include infection, aspergillosis, or persistent inflammation of the sinus floor [15,16]. In addition, the proximity of maxillary roots to the maxillary sinus is vital for endodontic microsurgical planning [17,18]. Hence, the current study aimed to evaluate the mean distance and differences between posterior maxillary teeth and maxillary sinus floor (MSF) concerning the age and gender of the patients.

## 2. Materials and Methods

The ethical clearance was obtained before starting the data collection from the Institutional Review Board, College of Dentistry, Prince Sattam Bin Abdulaziz University, Ministry of Higher Education, Kingdom of Saudi Arabia, with an ethical clearance number IRB PSAU2020022. The CBCT images were randomly selected, and the randomization process was started by generating numbers using a website https://www.calculator.net/random-number-generator.html (accessed on 10 January 2022). The number entered was 1, and for the number of records obtained from January 2016 to January 2019, we selected “generate 63 random numbers” on the website. Then, a list of random numbers appeared. This list of numbers is, one by one, cross-referenced with the patients’ file numbers based on the sequence of image acquisition; if a patient’s record does not follow the inclusion criteria, the record is excluded, and the following number on the list is used. For example, if the first number on the list is 4, this represents the fourth patient who acquired a CBCT image between January 2016 and January 2019, and so on.

In this cross-sectional study among the Saudi population, a total of 124 maxillary sinuses and 496 posterior maxillary teeth in 62 CBCT images were randomly selected, taken from a CBCT device CareStream CS 9300, the KV used was 90 KV, and the mA was 5.6 mA. The voxel size is 180–300 mm, and the field of view was 10 × 10 cm, examined using CS 3D imagining software (Carestream Dent L.L.C., Atlanta, GA, USA) and viewed on a 20-inch HP monitor (Hewlett-Packard, Palo Alto, CA, USA) working station desktop. The CBCT scans covering posterior maxillary teeth with fully formed teeth and nondistorted were included in the study, whereas scans of patients younger than 20, teeth with root resorption, teeth with periapical radiolucency, missing posterior teeth, and cases with large cysts or tumors were excluded from the current study.

### 2.1. Calibration

Calibration for the study was performed between the observer and an experienced endodontist. The observer was trained and calibrated for reading the CBCT images in a pilot study with a sample size of 10 (which was not included in this study). The observer evaluated the CBCT images using sagittal and coronal views to identify the relationship between the root apices of the posterior teeth and the maxillary sinus floor, and a single score was obtained for each tooth. A kappa reliability test on the observers was performed, and the obtained kappa value was 0.789. Disagreements were discussed, and a consensus was reached after the discussion.

The patients’ files were assessed by the Department of Oral & Maxillofacial Surgery and Diagnostic Sciences, College of Dentistry, Prince Sattam Bin Abdulaziz University, Al-Kharj, Saudi Arabia, to record the gender and the age of the subjects included in the study from September 2020 to December 2020, and further divided as male and female patients, depending on gender and age, into 4 groups (0 to 20, 21 to 40, 41 to 60, and above 60). Mean distances between posterior maxillary roots (PMRs) from different teeth and the maxillary sinus floor (MSF) were measured using a software-calibrated tool. Each measurement was obtained by measuring a line drawn from the MSF to the nearest point of the apex of the root in two views, sagittal and coronal, and the mean distance was calculated (Figure 1, Figure 2 and Figure 3). The distance for each root and each tooth were recorded and transferred to the Excel sheet.

### 2.2. Statistical Analysis

Data were recorded and evaluated using SPSS software (IBM, Chicago, IL, USA software version 26.0) for statistical analysis. Means and standard deviations were obtained from different measurements between the MSF and the root apices of posterior maxillary teeth. Independent *t*-test and one-way ANOVA were performed to see the relation between age and gender on these measurements. The *p*-value was set at α = 0.05.

## 3. Results

A total of 62 CBCT images were analyzed for root–MSF distance; 18 were women, and 44 were men, with a mean age of 29.63 (±10.6) years. Table 1 demonstrates that the mean root–MSF distances in the right and left first premolars are larger than the right and left second premolars. Table 2 summarizes the mean root–maxillary sinus distances of right and left first and second molars; the mesiobuccal roots of maxillary second molars frequently protrude within the maxillary sinus. Table 3 shows the measurements of distances between apices of the right premolar roots and the floor of the maxillary sinus concerning age and gender. No significant relation was found between the apices of right premolar roots and the floor of the maxillary sinus concerning gender and age, except for the age group 21–40 years, which showed a statistically significant relationship with the buccal root of the right first premolar (*p*-value < 0.05).

Table 4 describes the distances between the apices of the left premolar roots and the maxillary sinus floor concerning the age and gender of the patients. No significant relation was found between the apices of left premolar roots and the maxillary sinus floor concerning gender and age. Table 5 tabulates the measurements of distances between apices of the right first and second molar roots and floor of the maxillary sinus concerning age and gender. No significant relation was found between the apices of the right first and second molar roots and floor of the maxillary sinus concerning gender. A statistically significant association was found between the apices of right first and second molar roots and floor of the maxillary sinus concerning the age group 21–40 years (*p*-value < 0.05).

The measurements of distances between the left first and second molar roots and the floor of the maxillary sinus concerning age and gender are shown in Table 6. No significant relation was found between the apices of the left first and second molar roots and the floor of the maxillary sinus concerning gender. A statistically significant relation was found between the apices of the left first and second molar roots and the floor of the maxillary sinus concerning the age group 21–40 years (*p*-value < 0.05). 

## 4. Discussion

Previous experimental studies showed that the relationship between MSF and PMRs might affect various dental treatments and cause complications, such as maxillary sinus perforation or infection during extraction or root canal treatment, if the clinician does not understand this relationship and examine it through CBCT. Knowing the relationship between MSF and PMRs can help prevent odontogenic maxillary sinusitis [12].

During infancy, maxillary sinuses are filled with fluids. With growth, these fluids are absorbed, and the sinuses start to be filled with gas in a process called maxillary sinus gasification [13]. The sinus growth completion is rarely discussed in the literature. One study showed that sinus growth completion is around 18 years of age; however, the same study implied that it might continue due to the growth spurt’s remaining effect. Hence, our inclusion criteria were modified to include 20 and above to eliminate this variable [14]. This study aimed to assess the anatomical relationship of premolars, first and second molars, concerning the maxillary sinus using CBCT, and evaluate the age and gender effects on this relationship.

In Turkey, researchers looked at the anatomic relationship between maxillary third molars and maxillary sinuses and discovered that these teeth were frequently not in contact with the sinus floor [15]. Gu et al. investigated the association between the maxillary sinus and maxillary molars in China. The relationship between the posterior maxillary teeth and the MSF was split into three categories: type OS (the root apex extending below and outside the MSF), type CO (the MSF in contact with the root apex), and type IS (the root apex protruding into the MSF), and they mainly focused on the effect of age and missing teeth on the relationship [10]. Estrela et al. also assessed the relation between the MSF and the posterior maxillary teeth root apices. Similar to our results, it was noted that the shortest distance from the root apices to the MSF was the distance between the second molar mesiobuccal root and the MSF [9]. Pagin et al. also noted that the most frequent root in close relation to the MSF is the mesiobuccal root of the second molar [3]. Unlike our study, Kwak et al. revealed that the second molar’s distobuccal root was closest to the MSF in a study performed on a Korean population [17].

Mesiobuccal roots of maxillary second molars appeared to be the closest to the MSF. The frequency of root apices extending outside the MSF increased with age, while the frequency of the root apices extending inside or contacting with the MSF decreased with age. Similar to our results, Gu et al. concluded that the mesiobuccal roots of maxillary second molars are closer to the MSF compared to other teeth, and the chance of protruding inside the sinus decreases with age [10]. In the present study, no significant differences were reported in the proximity of roots between the right and left sides or between males and females. However, a study found a significant gender difference, indicating that males have roots closer to the sinus than females, but it did not discover any differences between the right and left sides [1,9]. Using CBCT in dentistry has many advantages, one of which is the assessment of the proximity of PMRs to the MSF, which is vital in endodontic procedures such as nonsurgical root canal treatment or endodontic microsurgery, especially if these operations are close to the sinus [8,17]. The current study focused on the AAE/AAOMR (2015) recommendations, and CBCT scans were screened for diagnosis, initial treatment of teeth with complicated morphology, or for the diagnosis and planning of endodontic disease after treatment. With the sample size taken from an endodontic clinic covering the whole spectrum of referral reasons, it replicated the real life-clinic situation in that the requirement for CBCT testing and modifications in treatment plans is also dependent on the patient’s history and other clinical findings. The strength of our cross-sectional study was that it was relatively quick and economical. We evaluated the prevalence and looked into the differences between posterior maxillary teeth and maxillary sinus floor (MSF) among the Saudi populations.

One of the limitations of this study is that the data source is derived from radiographic interpretation only and does not include other sources, such as dental history or any underlying chronic medical conditions. In addition, a relatively high statistical variance was observed, which was expected in this case. The data in this study are related to the location of an anatomical landmark with a degree of natural variation. Age, ethnicity, and gender might play a role in this variation. Other factors, including the history of sinusitis, trauma, genetics, and growth defects, also contribute to the sinus floor’s relation with the roots of posterior teeth. Some of these factors might affect any anatomical landmark regarding location, size, and relation with surrounding structures [18,19].

## 5. Conclusions

This study showed that the second molar mesiobuccal root apex is frequently related to the sinus floor. Differences were reported concerning age, concerning the distance between posterior maxillary teeth and the maxillary sinus floor. CBCT technology provided the clinical proximity between the MSF and the posterior teeth root apices during the treatment planning. The proximity of these anatomic structures should be considered to prevent an iatrogenic procedure and minimize the risk of an infectious disease within the sinus.

## Figures and Tables

**Figure 1 healthcare-11-00150-f001:**
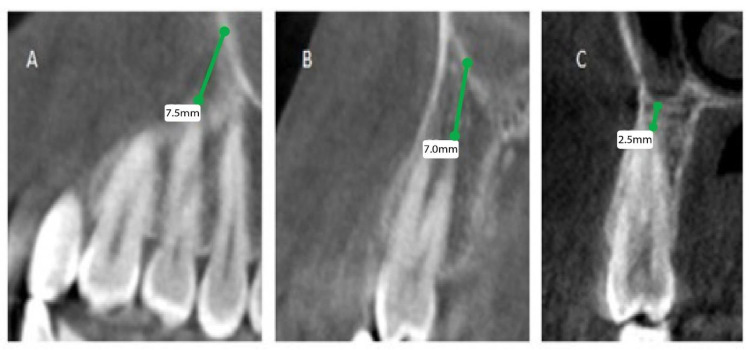
CBCT sagittal view showing the distance between the maxillary sinus and the root of permanent maxillary (right). (**A**) First premolar buccal root, (**B**) first premolar palatal root, (**C**) second premolar root.

**Figure 2 healthcare-11-00150-f002:**
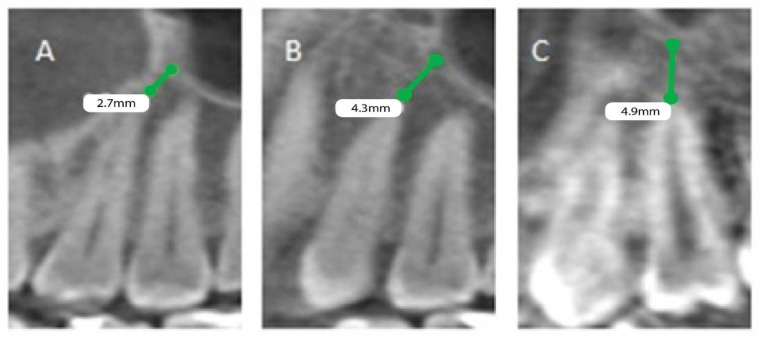
CBCT sagittal view showing the distance between the maxillary sinus and the root of permanent maxillary (left). (**A**) First premolar buccal root, (**B**) first premolar palatal root, (**C**) second premolar root.

**Figure 3 healthcare-11-00150-f003:**
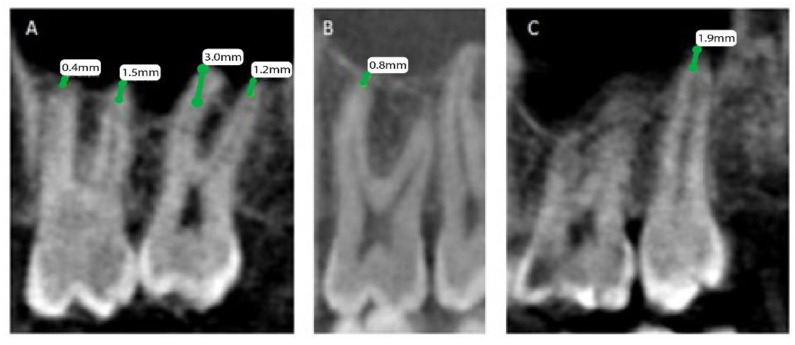
CBCT sagittal view showing the distance between the maxillary sinus and the root of permanent maxillary. (**A**) Right first and second molars with mesiobuccal and distobuccal roots, (**B**) left first molar mesiobuccal root, and (**C**) left second molar palatal root.

**Table 1 healthcare-11-00150-t001:** Mean root–maxillary sinus distances in mm for premolars.

	Right Side				Left Side			
	N	Mean ± SD	Min	Max	N	Mean ± SD	Min	Max
1st Premolar								
Buccal Root	62	6.11 ± 3.19	0.35	13.75	62	5.22 ± 3.41	−0.30	13.10
Palatal Root	49	5.06 ± 3.48	0.20	13.70	49	4.55 ± 3.56	0.00	13.05
2nd Premolar								
Root	62	2.43 ± 2.64	−1.85	9.65	62	2.08 ± 2.94	−2.60	13.35

**Table 2 healthcare-11-00150-t002:** Mean root–maxillary sinus distances in mm for 1st and 2nd molars.

	Right Side				Left Side			
	N	Mean ± SD	Min	Max	N	Mean ± SD	Min	Max
1st Molar								
Mesiobuccal Root	62	0.66 ± 2.72	−4.0	9.50	62	0.92 ± 2.69	−3.45	12.20
Mesiodistal Root	62	0.75 ± 2.43	−2.70	8.20	62	0.85 ± 2.68	−2.65	13.25
Palatal Root	62	0.69 ± 2.72	−4.95	935	62	0.48 ± 2.57	−3.40	7.85
2nd Molar								
Mesiobuccal Root	62	−0.24 ± 1.77	−3.15	6.45	61	−0.12 ± 1.94	−3.20	7.55
Mesiodistal Root	62	0.25 ± 1.95	−2.80	8.60	61	0.47 ± 2.07	−2.45	9.70
Palatal Root	62	0.59 ± 2.22	−4.40	9.65	61	0.79 ± 2.00	−2.75	6.45

**Table 3 healthcare-11-00150-t003:** Measurements of distances (mm) between apices of right premolar roots and floor of the maxillary sinus concerning age and gender (*n* = 62).

	14 Buccal Root			14 Palatal Root			15 Root		
	N	Mean ± SD	*p*-Value	N	Mean ± SD	*p*-Value	N	Mean ± SD	*p*-Value
Gender									
Male	44	5.53 ± 3.12	0.83 ^t^	37	4.66 ± 3.42	0.99 ^t^	44	2.02 ± 2.62	0.91 ^t^
Female	18	7.52 ± 2.99		12	6.28 ± 3.51		18	3.43 ± 2.50	
Age (years)									
0–20	9	4.31 ± 2.15		8	4.01 ± 2.20		9	2.10 ± 1.84	
21–40	46	6.03 ± 3.20	0.009 ^u^	35	4.88 ± 3.71	0.10 ^u^	46	2.25 ± 2.73	0.43 ^u^
41–60	6	8.09 ± 1.54		5	6.49 ± 0.98		6	4.01 ± 3.02	
>60	1	13.75 ± 0.0		1	12.45 ± 0.0		1	3.90 ± 0.0	

Significant level < 0.05; t: independent *t*-test; u: one-way ANOVA; 14: right 1st premolar; 15: right 2nd premolar; SD: standard deviation.

**Table 4 healthcare-11-00150-t004:** Measurements of distances (mm) between apices of left premolar roots and floor of the maxillary sinus concerning age and gender (*n* = 62).

	24 Buccal Root			24 Palatal Root			25 Root		
	N	Mean ± SD	*p*-Value	N	Mean ± SD	*p*-Value	N	Mean ± SD	*p*-Value
Gender									
Male	44	4.64 ± 3.28	0.49 ^t^	36	4.12 ± 3.37	0.22 ^t^	44	1.83 ± 3.08	0.69 ^t^
Female	18	6.63 ± 3.40		13	5.75 ± 3.93		18	2.68 ± 2.56	
Age (years)									
0–20	9	4.26 ± 2.79		8	4.34 ± 3.26		9	1.57 ± 2.20	
21–40	46	5.00 ± 3.36	0.06 ^u^	35	4.15 ± 3.51	0.09 ^u^	46	1.78 ± 2.73	0.10 ^u^
41–60	6	7.22 ± 3.31		5	6.15 ± 2.96		6	4.57 ± 4.47	
>60	1	12.30 ± 0.0		1	12.40 ± 0.0		1	5.20 ± 0.0	

Significant level < 0.05; t: independent *t*-test; u: one-way ANOVA; 24: left 1st premolar; 25: left 2nd premolar; SD: standard deviation.

**Table 5 healthcare-11-00150-t005:** Measurements of distances (mm) between apices of right 1st and 2nd molar roots and floor of the maxillary sinus concerning age and gender (*n* = 62).

	**16 Mesiobuccal Root**			**16 Mesiodistal Root**			**16 Palatal Root**		
	**N**	**Mean ± SD**	***p*-Value**	**N**	**Mean ± SD**	***p*-Value**	**N**	**Mean ± SD**	***p*-Value**
Gender									
Male	44	0.35 ± 2.70	0.88 ^t^	44	0.41 ± 2.37	0.76 ^t^	44	0.61 ± 2.78	0.98 ^t^
Female	18	1.43 ± 2.67		18	1.58 ± 2.43		18	0.89 ± 2.62	
Age (years)									
0–20	9	−0.52 ± 1.52		9	−0.20 ± 1.22		9	0.24 ± 0.65	
21–40	46	0.52 ± 2.63	0.02 ^u^	46	0.57 ± 2.25	0.008 ^u^	46	0.40 ± 2.60	0.03 ^u^
41–60	6	2.70 ± 3.43		6	2.43 ± 3.28		6	2.60 ± 3.99	
>60	1	5.75 ± 0.0		1	7.10 ± 0.0		1	6.75 ± 0.0	
	**17 Mesiobuccal Root**			**17 Mesiodistal Root**			**17 Palatal Root**		
	**N**	**Mean ± SD**	***p*-Value**	**N**	**Mean ± SD**	***p*-Value**	**N**	**Mean ± SD**	***p*-Value**
Gender									
Male	44	0.24 ± 2.18	0.36 ^t^	44	−0.27 ± 1.90	0.40 ^t^	43	0.49 ± 2.44	0.50 ^t^
Female	18	0.27 ± 1.90		18	−0.16 ± 1.46		18	0.81 ± 1.61	
Age (years)									
0–20	9	−0.68 ± 1.45		9	0.15 ± 1.05		9	0.13 ± 1.61	
21–40	46	−0.35 ± 1.45	0.001 ^u^	46	0.03 ± 1.52	0.001 ^u^	46	0.34 ± 1.68	0.001 ^u^
41–60	6	0.15 ± 2.54		6	0.15 ± 2.54		6	1.68 ± 3.45	
>60	1	6.45 ± 0.0		1	8.60 ± 0.0		1	9.65 ± 0.0	

Significant level < 0.05; t: independent *t*-test; u: one-way ANOVA; 16: right 1st molar; 17: right 2nd molar; SD: standard deviation.

**Table 6 healthcare-11-00150-t006:** Measurements of distances (mm) between apices of left 1st and 2nd molar roots and floor of the maxillary sinus concerning age and gender (*n* = 62).

	**26 Mesiobuccal Root**			**26 Mesiodistal Root**			**26 Palatal Root**		
	**N**	**Mean ± SD**	***p*-Value**	**N**	**Mean ± SD**	***p*-Value**	**N**	**Mean ± SD**	***p*-Value**
Gender									
Male	44	0.79 ± 2.90	0.52 ^t^	44	0.75 ± 3.02	0.16 ^t^	44	0.36 ± 2.59	0.77 ^t^
Female	18	1.33 ± 2.11		18	1.12 ± 1.64		18	0.79 ± 2.58	
Age (years)									
0–20	9	0.34 ± 1.14		9	0.11 ± 1.32		9	0.92 ± 1.77	
21–40	46	0.68 ± 2.25	0.007 ^u^	46	0.55 ± 1.95	0.001 ^u^	46	0.18 ± 2.50	0.03 ^u^
41–60	6	2.27 ± 4.94		6	2.75 ± 5.20		6	0.95 ± 2.68	
>60	1	9.05 ± 0.0		1	10.25 ± 0.0		1	7.50 ± 0.0	
	**27 Mesiobuccal Root**			**27 Mesiodistal Root**			**27 Palatal Root**		
	**N**	**Mean ± SD**	***p*-Value**	**N**	**Mean ± SD**	***p*-Value**	**N**	**Mean ± SD**	***p*-Value**
Gender									
Male	43	−0.26 ± 2.08	0.60 ^t^	43	0.40 ± 2.32	0.29 ^t^	43	0.67 ± 2.12	0.37 ^t^
Female	18	0.22 ± 1.56		18	0.61 ± 1.36		18	1.05 ± 1.73	
Age (years)									
0–20	9	−0.96 ± 1.42		9	−0.03 ± 1.20		9	0.52 ± 1.93	
21–40	45	−0.20 ± 1.50	0.006 ^u^	45	0.21 ± 1.37	0.001 ^u^	45	0.71 ± 1.83	0.03 ^u^
41–60	6	0.80 ± 3.65		5	1.81 ± 4.06		6	0.78 ± 2.49	
>60	1	5.55 ± 0.0		1	8.60 ± 0.0		1	6.45 ± 0.0	

Significant level < 0.05; t: independent *t*-test; u: one-way ANOVA; 26: left 1st molar; 27: left 2nd molar; SD: standard deviation.

## Data Availability

Any data related to the study can be provided upon a reasonable request.

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
