# Peer review of "Anatomical Evaluation of Posterior Maxillary Roots in Relation to the Maxillary Sinus Floor in a Saudi Sub-Population: A Cross-Sectional Cone-Beam Computed Tomography Study"

_healthcare, 2023, doi:10.3390/healthcare11010150_

Round 1
Reviewer 1 Report
“Anatomical evaluation of posterior maxillary roots in relation to the maxillary sinus floor in a Saudi sub-population: A CBCT study” was submitted to Healthcare
This study aimed to evaluate the mean distance between posterior maxillary teeth and maxillary sinus floor and determine the gender and age regarding the distance. The authors concluded that the second molar mesiobuccal root apex is frequently related to the sinus floor.
The manuscript deals with an interesting issue; however, there are several concerns related to the study.
The entire manuscript requires language review.
Title: Please include the study design. Please define CBCT
Abstract
-. Objective: It is not clear if age and gender are really determined considering the distance.
-. Methods: Please define CBCT. Although it is obvious, it is important to indicate that the 62 images corresponding to 62 patients.
-. Results: lines 25-26. p-value must be presented. Nothing is indicated regarding gender.
-. Conclusions: Considering the objectives, nothing is concluded with concerning age and gender.
Keywords: Endodontic microsurgical planning and posterior maxillary roots are not MeSH terms.
Introduction
-. Line 50. Please do not start a sentence with a number (10-12%). Indicate the population from which this information comes.
-. Line 53. Reference 7 does not correspond to Maillet et al.
-. The novelty of the study is questionable; for this reason, the contribution that this work carried out in Saudi Arabia brings to knowledge should be indicated. How could you differentiate it from other populations that have explored the same topic? (Major concern). What is its external validity? Please comment on this.
-. Lines 82-85. The goals are very difficult to understand. Requires a language edition.
Methods
-. Line 90. The word "descriptive" is unnecessary.
-. Line 91. Please present the randomization process in detail.
-. Line 97. Capitalized initials of some words should be corrected.
-. Page 3. Line 102. Please present the results of the calibration and the statistical test used.
-. Page 3. Line 113. Please indicate the theoretical/scientific foundation that supports the stratification of the formed age groups. The image quality of all figures should be improved.
-. Page 4. Line 132. Please indicate the statistical test used to establish the normal distribution of the data, and its result. Was the sample size calculated?
Ethical considerations: This reviewer does not understand why the dean authorized the assessment of patient data. Did the patients sign informed consent? If so, was the authorization to obtain that information described in the document?
Results
-. The obvious greater number of men is striking. Please comment on this and its implications on the results.
-. The largest number of patients in the 21-40 stratum is evident. It is possible that the small number of patients, in the other strata, detracted from the power of the sample to find significance (major concern). Please comment on this.
Discussion.
-. Many observations described above should be discussed in this section.
-. Page 8, lines 217-220. This information is unrelated to the study findings.
-. Page 8. Line 226. Since this study did not address ethnicity, information on ethnicity should be provided. Moreover, the current investigation did not describe any information related to gender.
-. Page 8. Lines 226-229. Present the references that support this information.
Conclusions
The conclusions are not accounting for all the established objectives (relationship with age and gender).
Author Response
We would like to thank the academic editor and the reviewers for taking out their immensely valuable time to review this manuscript and give us their comments. We would like to explicitly state that we agree with all the comments as these helped us improve the quality of our paper. We have made a conscious effort to answer all the remarks in the paper as advised by the reviewers and highlighted changes made in red for their convenience. Kindly consider these and excuse us for any lapse on our part.

Reviewer 2 Report
Dear authors, thank you very much for submitting your study to the journal.
Here goes a few concerns:
Please notice that according to the authors guidelines, the Abstract does not require subheadings
The extended name for CBCT should appear both in the Abstract and body of manuscript.
The Introduction looks fine
How were the patients selected?
What was the sampling method? All included? Any specific chart numbering? Any other?
The Title mentions Saudi patients. But it is not clear in the methods if only Saudi patients were recruited.
Any information regarding the CBCT mA and Kv?
Why were the CBCT examinations conducted? Not because of the study, right?
How many observers conducted the measurements? Did the authors conduct any reliability test on the Observer?
In the Tables 1, 2 and 3 is not clear what are the units to be considered. Was that mm?
I suggest the authors to debate in the Discussion the recommendations for making a CBCT imaging according to the AAE, since the Diagnosis is one of them.
The study strength should be addressed.
The external validity and results generalization should be debated
Any recommendation for further studies?
Please notice that the references are not according to the journal guidelines.
Author Response

(The authors gave the same response as above.)

Round 2
Reviewer 1 Report
The authors made the suggested corrections. Publication of the manuscript is recommended
Reviewer 2 Report
Dear authors, I have no more concerns.